

# The impact of peer review on the contribution potential of scientific papers

Akira Matsui[1,2,*], Emily Chen[1,2,*], Yunwen Wang[3] and Emilio Ferrara[1,2,3]

[1] Department of Computer Science, University of Southern California, Los Angeles, California, United States
[2] Information Sciences Institute, University of Southern California, Marina del Rey, California, United States
[3] Annenberg School for Communication and Journalism, University of Southern California, Los Angeles, California, United States
* These authors contributed equally to this work.

## ABSTRACT

The peer-reviewing process has long been regarded as an indispensable tool in ensuring the quality of a scientific publication. While previous studies have tried to understand the process as a whole, not much effort has been devoted to investigating the determinants and impacts of the content of the peer review itself. This study leverages open data from nearly 5,000 PeerJ publications that were eventually accepted. Using sentiment analysis, Latent Dirichlet Allocation (LDA) topic modeling, mixed linear regression models, and logit regression models, we examine how the peer-reviewing process influences the acceptance timeline and contribution potential of manuscripts, and what modifications were typically made to manuscripts prior to publication. In an open review paradigm, our findings indicate that peer reviewers' choice to reveal their names in lieu of remaining anonymous may be associated with more positive sentiment in their review, implying possible social pressure from name association. We also conduct a taxonomy of the manuscript modifications during a revision, studying the words added in response to peer reviewer feedback. This study provides insights into the content of peer reviews and the subsequent modifications authors make to their manuscripts.

Corresponding author
Akira Matsui, amatsui@usc.edu

## INTRODUCTION

Although the peer-reviewing process often feels like a heavy burden to many scientists (*Djupe, 2015*; *Golden & Schultz, 2012*; *Kovanis et al., 2016*), the process itself plays an integral role in scientific research. During the review process, reviewers are sometimes asked to decide whether to accept, request a major or minor revision or reject a submitted paper. It is well known that this process enables the quality of published scientific papers to be maintained to a certain extent (*Bornmann, 2011*). Reviewers are also able to leave feedback, which enables authors to revise the paper to improve upon or extend the paper. While the scientific community generally believes that this peer-review process gives credence to the competency of a published paper (*Hemlin & Rasmussen, 2006*), we do not necessarily understand if this process improves the contribution potential of a paper.
We define contribution potential as the subsequent contribution and impact that a paper amasses over time (see Definition 1). In this paper, our working definition of the contribution potential of a given paper is how much recognition and engagement a paper has received, which we estimate as a paper's citation count and its citation metrics. Specifically, a paper's citation count refers to the number of citations the paper receives, and its citation metrics refer to Altmetrics and the readership numbers in our study. We provide more detail on these measures in "Data".

**Definition 1 (Contribution Potential)** *The contribution potential of a given paper measures how much recognition and engagement an article has currently received. We use the (1) number of citations, (2) Altmetrics and (3) number of readers as proxies for a paper's future potential to contribute to its field. We refer to (2) and (3) collectively as a paper's citation metrics.*

We focus on the concept of "science of science"—the study of how science and research is conducted—due to significant implications of potential reduced efficiency with the limited resources of the scientific community (*Fortunato et al., 2018*). If the peer-review process merely checks that the quality and potential for contribution meets a certain threshold, rather than striving to improve the submitted work, we then question the effectiveness of the current review process. We can only ascertain whether the current system needs to be rectified if we first study the process itself. In this particular study, we investigate the effect of peer review content, author revisions, author rebuttals and their consequences on a paper's contribution potential. Although reviewers may make suggestions on methods or analyses for increasing research validity, it should be noted that the role of peer review more often lies in improving the scientific reporting. We recognize the value of the latter scenario, as the quality of reporting is an important attribute that adds to the quality and rigor of the work itself. In reality, the peer review process can take on a combination of both roles. To understand the nature of peer review, we seek to contextualize the direct impact of peer review on a scientific work by comparing the work's content before and after peer review, *e.g.*, by examining manuscript modifications made in each review iteration.

In this study, we aim to use objective measurements for assessing peer review and understanding how the peer-review process may improve the submitted manuscripts by asking the following research questions:

**RQ1:** What factors from the revision process predict the acceptance timeline and contribution potential of a submitted manuscript?

**RQ2:** To what extent is peer review sentiment associated with author or reviewer characteristics?

**RQ3:** During the revision process, what kinds of modifications do authors make to their submitted papers?

We note here that peer review data for papers that were eventually rejected are not publicly accessible and are hence not in our current dataset. This limits the scope of our study to focus on the peer review process and contribution potentials of accepted and published papers—we hope to expand our dataset to include rejected paper submissions in future expansions of this work.

## LITERATURE REVIEW

The Internet has brought the peer-reviewing process to various online platforms such as Easychair (https://easychair.org/). It also accelerates the exchange of reviewer feedback and author rebuttals. The transition from paper-backed peer reviews to electronic peer reviews produces an interactive publication and open peer commentary (*Harnad, 1996*). The open review paradigm also provides the research community easier and arguably more equal access to peer reviews, which allows researchers to dive into the content of reviews. *Grimaldo, Marušić & Squazzoni (2018)* conducted a quantitative analysis of publications from 1969 to 2015 and found a large increase in the number of publications that examine peer review. However, they noted that these works constituted a small portion of the entire scientific literature landscape, *e.g.* 1.8 pieces in every 10,000 articles in 2015. Among these publications on peer review, only half were empirical research articles. The gaps in current research present a need for more empirical research that can leverage the advantages of open review to study peer review impact.

Research on peer review, to date, has expressed a variety of concerns about the quality of reviews (*Henderson, 2010*; *Schroter et al., 2004*), which consequently leads to the questioning of the effectiveness of peer reviewing itself (*Henderson, 2010*). Previous experimental studies have tried to improve the quality of peer reviews through reviewer training (*Schroter et al., 2004*) but improvement was slim. There is a lack of well-recognized standards when attempting to evaluate the quality of peer reviews. For example, *Justice et al. (1998)* instructed manuscript authors and editors to judge review quality on a five-point Likert scale; *Wicherts (2016)* posits the transparency of the peer review process as a proxy for the quality and developed a 14-item scale to have raters subjectively measure transparency. According to a methodological systematic review by *Superchi et al. (2019)*, there have been 23 scales and one checklist used to assess the quality of peer review reports. As many as 25% of these tools simply use a single item, and the rest have four to 26 items, all relying on self-reporting. The tools in existing literature do not address the concerns that (1) subjective measurement may not be reliable and (2) the evaluation of peer review efficacy may be concerned with more factors beyond the transparency suggested by *Wicherts (2016)* or other single dimensions of the peer review process. The contribution potential of a paper has many facets, and there are many ways to operationalize its measurement. In our study, we measure "contribution potential" as the citation metrics and a paper's citation count (see Definition 1).

More recent research has also tried to understand the peer review process as a whole through exploring biases (*Tomkins, Zhang & Heavlin, 2017*), gender (*Card et al., 2020*), and effects of the open review system on peer review (*Bravo et al., 2019*). Apart from the quality of a paper, there exist numerous variables about a paper, its author(s), and its reviewer(s) that can influence the substance of the peer review a paper receives. Previous meta-science works present similar or conflicting findings on the reviewer responses to the authors' gender (*Grogan, 2019*; *Murray et al., 2019*; *Laycock & Bailey, 2019*), racial diversity and seniority on publication outcomes such as acceptance rates or paper citations. While the scientific community generally perceives gender and racial diversity as a benefit

to scientific advancement (*Nielsen et al., 2017*), there may be evidence indicating that female researchers are under-represented (*Hechtman et al., 2018*). Recently, (*Grogan, 2019*) reported a lower acceptance rate of papers with a female as a last or corresponding author in science, technology, engineering and mathematics (STEM) fields, compared to papers with a male last or corresponding author. Meanwhile, manuscripts with male last authors are 7% more likely than those with female last authors to be accepted; this gender effect increases acceptance likelihood even more when all authors are male, compared to the acceptance rates of a mixed-gendered team (*Murray et al., 2019*). However, some researchers showed that papers in which the first author is female are actually accepted more than papers in which the first author is male (*Laycock & Bailey, 2019*). In an open journal setting where author information is publicly available, studying peer reviewers' responses to a paper in accordance to the author gender may be pertinent to assessing the equity of the review process.

In a recent attempt to assess peer review biases and robustness, *Buljan et al. (2020)* operationalized "review robustness" as the lack of influence of extrinsic variables (*e.g.*, area of research, type of peer review, and reviewer gender) on linguistic characteristics of the peer review. They tested for and established that "review robustness" exists in the 583,365 peer review reports from 61 journal published by Elsevier, constituting perhaps the largest-scale study on peer review to date. However, one limitation of *Buljan et al. (2020)* is a consequence of its large sample size. Despite their control of the papers' four research areas, the study failed to fully consider the statistical challenge with different data distributions and variances across so many journals. Our study focuses on one specific journal, PeerJ, to control for any journal-to-journal confounders, without excessively extrapolating findings to the whole scientific community.

## DATA

We leverage PeerJ (https://peerj.com/) as our primary data source and supplement this with publication performance and contribution data from Altmetrics (https://www.altmetric.com/), Crossref (https://www.crossref.org/) and Dimensions (https://www.dimensions.ai/) (*Orduña-Malea & Delgado-López-Cózar, 2018*). We detail the data extraction and pre-processing methods in the following sections.

### PeerJ data

PeerJ is an open access journal that has a single-blind review process during the pre-publication process, but post-publication, allows the authors of published papers the option to publicize the paper's review history. Reviewers are also able to "sign" or associate their names with their reviews after the paper has been accepted for publication. We note that unlike many other journals, PeerJ asks the reviewers to evaluate papers based on an objective determination of scientific and methodological soundness, rather than subjective evaluations of "impact", "novelty" or "interest". All peer reviews are conducted by research scientists typically invited by the editor(s) of the journal, and final decisions on whether to reject, request a minor or major revision, or accept the paper are made by

**Table 1 PeerJ journal and number of articles published per journal as of October 30, 2019.**

| PeerJ Journal Research Areas | # of Articles Published | # of Articles Published w/Open Reviews |
|---|---|---|
| Life & Environmental | 7,912 | 4,917 (62% of the total) |
| Computer Science | 277 | 165 (60% of the total) |
| Physical Chemistry | 5 | |
| Organic Chemistry | 1 | |
| Inorganic Chemistry | 0 | |
| Analytical Chemistry | 1 | |
| Materials Science | 2 | |

the editor(s). PeerJ is also included in the Web of Science database (https://clarivate.com/webofsciencegroup/), a well known research citation database, which checks to ensure that the journal satisfies certain criteria such that Web of Science acknowledges the journal as a scientific journal and not as a predatory publisher (*Bowman, 2014*; *Clark & Smith, 2015*).

PeerJ covers a wide range of research disciplines, mostly within the natural sciences (see Tables 1 and 2 for the full list of research areas and the number of articles per journal which have opted to make their review histories public). While they do publish in numerous areas, as of October 30, 2019, the majority of the articles with open reviews are published within *PeerJ—the Journal of Life and Environmental Sciences* and *PeerJ Computer Science*. Because the number of articles published in PeerJ's chemistry and material science journals are relatively low, we exclude these articles from our dataset, and focus on articles published in *PeerJ—the Journal of Life and Environmental Sciences* and *PeerJ Computer Science*.

## Data collection and pre-processing

We were able to crawl PeerJ's website for the articles that opted to make their review histories public, which resulted in a 62% yield of articles in the *PeerJ—Journal of Life and Environmental Sciences* and a 60% yield in *PeerJ Computer Science*. We collected this data on October 30, 2019 and the number of articles with available review histories may have changed since initial data gathering due to additional articles being published. From the articles with open review histories, we then found the article's associated peer reviews, author rebuttals, and paper revision histories. Some peer reviews, rebuttals, and revision histories may be missing from the dataset due to download failures. PeerJ only includes the review histories alongside published works, which means that we are unable to collect review histories from papers that were eventually rejected by the editor. While PeerJ does leverage different methods to invite reviewers to review a manuscript (*e.g.* volunteering, editor invitation), we are unable to differentiate solicitation modality through the retrieved data.

Each article consists of at least one revision and subsequent article version. The versions of the articles begin at 0.1—indicating the original submission and the first-round of peer review—and increments the version number by 0.1 with each following review round.

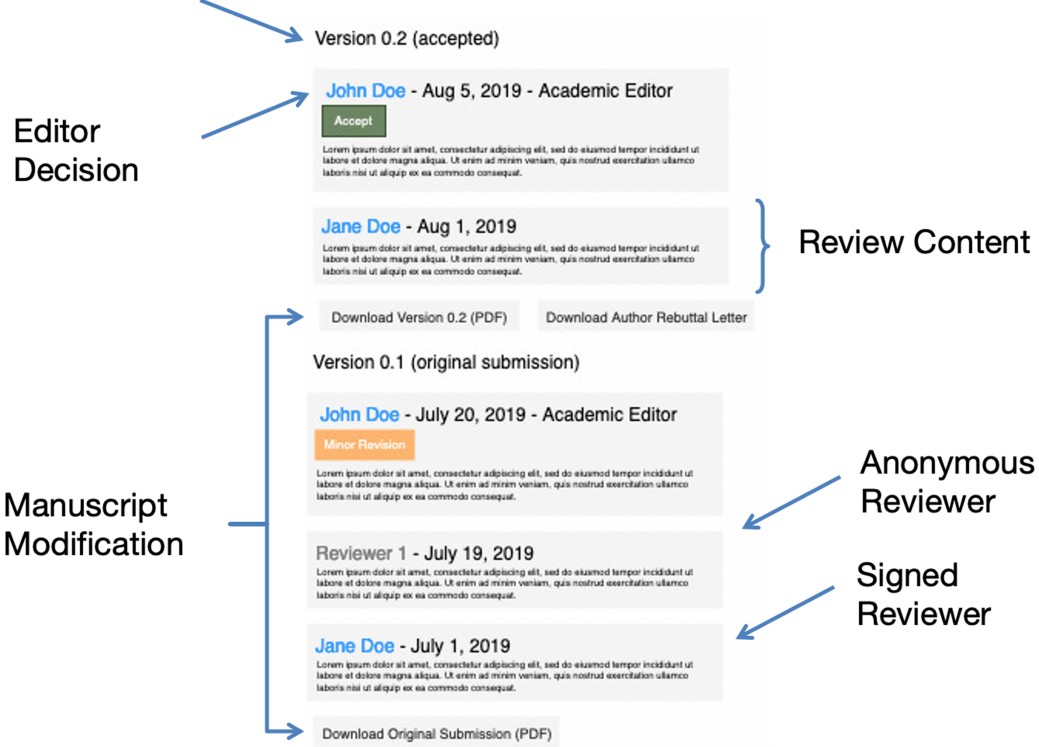

**Figure 1  The structure of the PeerJ open review audit histories, with labeled areas of interest.**

For each revision version, we determined if a peer reviewer has "signed" his or her review. In addition to peer reviewers' comments on the manuscript, we also extracted the editor's comments and decisions as well as the article's DOI (Digital Object Identifier). See Fig. 1 for the PeerJ open review audit history structure.

### Peer reviews

Each peer reviewer is asked to comment on the assigned article in the following aspects:

1. Basic Reporting
2. Experimental Design
3. Validity of the Findings
4. Comments for the Author

All feedback for the authors are contained to these questions, with many of the miscellaneous and specific comments from peer reviewers being enumerated in the "Comments for the Author" section. Because the review audit trails follow the same structure in both PeerJ journals, we are able to scale and parse all available peer reviews, and scrape the peer review data directly from the PeerJ article HTML.

**Table 2 Earliest and latest article publication dates from PeerJ's respective journal research area with available review histories as of October 30, 2019.**

| PeerJ Journal Research Areas | Earliest Article Published | Latest Article Published |
| --- | --- | --- |
| Life & Environmental | February 12, 2013 | October 18, 2019 |
| Computer Science | May 27, 2015 | October 28, 2019 |

### Author rebuttals

We downloaded the authors' rebuttals or response letters addressing peer reviewers' comments, which were mostly in docx format, from an article's HTML page, and extracted the raw text using Python's docx library. We excluded all rebuttals that were not provided in docx format for the sake of scalable processing. Because the first version of the article is the initial submission, version 0.1 will not have a rebuttal. In total, we found 6,149 rebuttals associated with the articles included in this study, with 6,049 rebuttals in *PeerJ—Journal of Life and Environmental Sciences* and 100 rebuttals in *PeerJ Computer Science*.

### Submission versions

We also wanted to track how the papers evolved through revision iterations between the authors and the peer reviews. PeerJ's open review history provides the updated article that the authors submit for the next round of peer reviews, and this enabled us to download each iteration of the paper, including the original submission and the final submission to be published. These submission versions are uploaded as PDFs, so we used the Python library PyPDF2 (https://github.com/mstamy2/PyPDF2) for text extraction. Reading the raw text from the PDFs was not as clean as we had hoped, due to variance in encoding and metadata that is included with different PDFs. In future work, we hope to first convert these PDFs into Word documents for cleaner text extraction.

Once we obtained all submission versions for a particular article, we found the differences in text between contiguous submission versions only (*e.g.* we found the differences between version 0.1 and version 0.2, and between version 0.2 and version 0.3). In order to find these differences, we counted the individual occurrences of words for each article version. This approach then allowed us to find the words that are added or removed from version to version. We do note here that we lost the ordering of the words, but as we leveraged Latent Dirichlet Allocation (LDA) topic modeling, which turns a document into a bag of words, the ordering did not impact our results *Blei, Ng & Jordan (2003)*. For each set of differences, we identified two groups of words—the first group being the words "added" (*i.e.* present in the word count dictionary of the current revision but not the previous revision), and the second group being the words "removed" (*i.e.* not present in the word count dictionary of the current revision but present in the prevision version). In this paper, we focus only on the words in the first revision that were "added" to the original manuscript.

We generated 7,587 submission differences from the *PeerJ—Journal of Life and Environmental Sciences* and 234 submission differences from *PeerJ Computer Science*.

*Article metric collection*

We collected article metrics using Dimensions, which is a service that provides organized scholarly information (*Orduña-Malea & Delgado-López-Cózar, 2018*). Dimensions uses Crossref internally to help construct their article information database. The Dimensions API (Application Programming Interface) allowed us to extract comprehensive information on a particular scholarly article, such as citation statistics, grant information, and Altmetrics. We also leveraged the API to find the h-index for the authors in our PeerJ dataset. Additionally, we used publicly available Altmetric scores and the summed number of readers from reference manager applications (Mendeley, Connotea and Citeulike), which were retrieved through the Altmetric API (https://api.altmetric.com/), to calculate metrics for the contribution potential of papers in our dataset.

# METHODS

*Peer review attributes of interest*

We study the impact the peer reviewing process has on a published article's contribution potential, as defined in Definition 1, by examining four attributes from the publishing process:

1. peer review content
2. number of review iterations
3. peer review editor decision
4. manuscript modifications made in each review iteration

We also summarize the variables used in our analysis in Table A3 in the Appendix.

## Peer review content

We focus on the sentiment of review content. We leverage PeerJ's single blind review process and the optional post-publication ability for reviewers to "sign" their reviews to compare the sentiment of peer review feedback from reviewers who opt to "sign" their reviews and from those who choose to remain anonymous.

## Review iterations

Each submitted manuscript goes through at least one round of revision with all of the peer reviewers, if not more, prior to publication. As the revision process between the paper's authors and the peer reviewers is intended to help check and improve the reporting quality and completeness of the final publication, we investigate whether the number of review rounds affects the paper's eventual contribution potential.

## Peer review editor decision

After all peer reviewers have submitted their feedback for a particular revision iteration, the PeerJ editor makes a final decision on whether to reject, request minor or major revisions, or accept the paper for publication. This decision is made by the editor after taking the peer reviewer's feedback into consideration, and assessing the collective opinion

on the paper's quality. We note that all of the papers in our dataset have been eventually "accepted", as the "rejected" papers are not published.

However, external factors may play a role in the editor's final decision. Since PeerJ uses a single blind review process, the authors' genders (inferred by name), academic prestige or prior experience may potentially influence the verdict. For example, manuscripts written by prestigious authors may be accepted with fewer revisions, while manuscripts written by emerging authors may elicit more requests for revisions.

## Manuscript modifications

If the editor has requested a minor or major revision, authors are given the opportunity to incorporate the peer reviewers' feedback into their work and submit a new iteration of their manuscript. Some of these changes may be to simply rectify grammar while others change the implications of the paper. Identifying the specific changes made to the revised submissions enables us to understand how the paper content evolves with peer reviewer input.

## Prediction of a paper's acceptance timeline

We study if author gender or reputation impacts the editor's decision to request a revision (minor or major) or accept the paper. We use Genderize.io (https://genderize.io/) to predict the author gender based on the author's first name. Genderize.io has been used in prior research and commercial settings to estimate gender (*Santamaría & Mihaljević, 2018*). We define three variables to describe the gender constitution of the authors of a specific article: the gender of the first author, the gender of the last author, and the percentage of female authors out of all of the paper's authors[1].

We use the h-index of the author as a proxy to quantify his or her prestige or reputation in academia. The h-index is a well-known metric that a given researcher has, where the researcher has published $h$ research articles and each of the $h$ papers has been cited at least $h$ times. We use the h-index to our advantage in this study, as it takes into account all of the author's published papers, instead of a singular paper (*Hirsch, 2005*). As a result, we use the authors' h-index with the assumption that the h-index is not defined only by the number of citations the article in our dataset receives. We use the average h-index of all of the authors and the maximum h-index out of all the authors in a particular manuscript. Due to API and data limitations, we are unable to retrieve the authors' h-index when the paper was reviewed. Thus, we use the current h-index of the authors. As the peer reviewer's comments may not accurately reflect the contribution potential of the article, we use the h-index to capture this omitted factor and use the number of citations for the article to test the robustness of our results.

We also infer reviewers' perception and attitude towards the assigned manuscript through the polarity of their review. The authors' response to the peer reviewers' feedback might also affect the reviewers' view of the paper. Therefore, we use the polarity of the authors' rebuttal as a control variable. To estimate the polarity of a given text, we use TextBlob (https://textblob.readthedocs.io/en/dev/). TextBlob returns a polarity score

[1] The authors of this paper recognize that the gender is not binary. Our representation of the gender information does not reflect our attitude towards any gender
within the range (−1, 1), where negative values mean negative sentiment and positive values correspond to positive sentiment.

We acknowledge that the review process may differ depending on the paper's research area, with some research areas preferring to request further revisions and others satisfied with a quick review process. Due to the potential heterogeneity in the decision processes, we use a conditional mixed logit model to allow us to incorporate a fixed effect in the model (*Greene, 2003*). We run this model on the editor's decision, incorporating the fixed effect for each research area with the following specification:

$$y_i = \mathbf{x}_i'\beta + \mathbf{z}_i'\gamma + \varepsilon_i \tag{1}$$

where $y_i$ is a binary variable that is 1 if the editor decides to accept article $i$ for publication and 0 if a revision is requested, and where $\varepsilon_i$ is the error term. $\mathbf{x}_i$ are the vectors that describe the variables not related to the reviewed manuscript (*e.g.* gender and prestige) and $\beta$ is the parameter of interest. $\mathbf{z}_i$ are the vectors for the polarity control variable discussed above. For a robustness check, we estimate $\beta$ with different $\mathbf{x}_i$ vectors, changing variables for gender information or h-index values.

## Prediction of contribution potential

The editor requests that the authors submit a revised version of their publication in order to ensure that the discrepancies found by the peer reviewers and the editor are addressed appropriately. The peer reviewers and the editor have a responsibility to verify that all published papers satisfy the journal's publication criteria, and reject those that do not meet this expectation. The revision process enables the authors and the peer reviewers to have a dialogue on how to improve the submitted paper so that it meets these requirements.

We use a linear mixed model to determine if the number of revisions is correlated with the number of citations. We show that the number of revisions is not correlated with the paper's contribution potential, but we note here that as we were unable to obtain a dataset that includes the revision history of rejected papers, our conclusions only hold true for accepted and published manuscripts. The relationship between independent variables and citation number can vary significantly based on an article's research area, and we use a mixed linear effect model to account for this potential heterogeneity. Although random effects may differ by field, the fixed effect is common for all research areas.

We control for confounding variables that might affect the citation number. We specifically control for the research area, length of time since publication, number of authors and authors' h-index. The citation count of a particular publication may depend on the publication's field, and we control for this heterogeneity as the random intercept for each research area. We expect the number of citations a publication receives to increase over time and control for this by finding the number of months that have elapsed since publication. To control for this, we use the variable Year-month, which is a linear trend variable to control for differences in citation counts depending on when the paper was published. We control for the number of authors on a paper, as more contributing authors can increase the exposure and, consequently, citation count of a

paper. We also use the h-index of the authors to control for the impact that an author's visibility in the field might have on citation count.

## Review content sentiment

Sentiment analysis of peer reviewers' feedback allows us to study (1) if author h-index affects the sentiment of the reviews, and (2) if there is a difference in sentiment between reviewers who choose to "sign" their reviews and those who choose to remain anonymous. We again use TextBlob to calculate the polarity and subjectivity of the reviews for each reviewer. While the polarity is mapped onto (−1, 1) to reflect polarity sentiment, the returned subjectivity value ranges from (0, 1), with a higher value corresponding to higher subjectivity. We also use the authors' inferred genders and h-indexes in our model. We leverage the cross term of the female author ratio and if the reviewer chose to remain anonymous to study whether the relationship between the authors' gender and the reviewer's sentiment differs depending on whether or not the reviewer "signs" the review.

To test our hypothesis, we use a mixed linear regression model, which enables us to estimate the fixed effects across all reviews. Sentiment towards the submitted manuscript can depend on the article content and on the peer reviewer. By using the mixed linear effects model, we can take this into account and decompose the coefficient of the independent variables into fixed and random effects. In this analysis, the fixed effect can be interpreted as the general tendency across reviews for all reviews.

## Taxonomy of manuscript modifications during a revision

During the review process, authors are able to revise their manuscripts and incorporate the peer reviewers' feedback and suggestions into the next iteration of their paper. Some authors even self-identify areas in their papers to improve upon and make corresponding changes for the next revision submission. This modification process is integral to the review process as it enables constructive discussion and improvement upon papers prior to publication.

We aim to understand what kinds of changes are made when authors revise their manuscripts. In order to do so, we conduct topic modeling with Latent Dirichlet Allocation (LDA) on the words that were added in a new revision of the manuscript. LDA is a statistical tool that identifies latent topics in a given *corpus* of documents (*Blei, Ng & Jordan, 2003*). In this study, we focus specifically on words *added* in a revision and on the changes made between the original submission and the first revision. We define a document as the words that are added in the revision of a particular manuscript, which is then represented as a bag-of-words (BoW). To mitigate the impact of specific research topics or terminology in our topic modeling work, we also removed the words that appear in less than fifty documents (about 1% of the entire documents, 50/4,899 ≈ 1%).

## RESULTS

In this section, we present the results of our analyses to answer the research questions proposed in the Introduction. We first discuss the results of our regression model to predict the acceptance timeline of a paper based on attributes of the first revision

**Table 3 Conditional logit models: revision decision.**

| | Accepted at the first revision | | | | |
| | (Model 1) | (Model 2) | (Model 3) | (Model 4) | (Model 5) |
|---|---|---|---|---|---|
| Maximum h-index | 0.005*** | | | | |
| | (0.001, 1.0050) | | | | |
| | [0.0–0.01] | | | | |
| Average h-index | | 0.012*** | 0.012*** | 0.012*** | 0.012*** |
| | | (0.002, 1.0121) | (0.002, 1.0121) | (0.002, 1.0121) | (0.002, 1.0121) |
| | | [0.01–0.02] | [0.01–0.02] | [0.01–0.02] | [0.01–0.02] |
| Female last author | | | −0.045 | −0.044 | −0.044 |
| | | | (0.057, 0.9560) | (0.057, 0.9570) | (0.057, 0.9570) |
| | | | [−0.07 to 0.16] | [−0.07 to 0.16] | [−0.07 to 0.16] |
| Female first author | | | −0.004 | −0.001 | 0.0004 |
| | | | (0.053, 0.9960) | (0.053, 0.9990) | (0.053, 0.9960) |
| | | | [−0.1 to 0.11] | [−0.1 to 0.1] | [−0.1 to 0.1] |
| Female author ratio | | | 0.038 | 0.036 | 0.035 |
| | | | (0.114, 1.0387) | (0.114, 1.0367) | (0.114, 1.0356) |
| | | | [−0.19 to 0.26] | [−0.19 to 0.26] | [−0.19 to 0.26] |
| Polarity of authors' rebuttal | 0.146 | 0.179 | 0.183 | 0.190 | 0.207 |
| | (0.251, 1.1572) | (0.251, 1.1960) | (0.251, 1.0387) | (0.251, 1.2092) | (0.251, 1.2300) |
| | [−0.35 to 0.64] | [−0.31 to 0.67] | [−0.31 to 0.67] | [−0.3 to 0.68] | [−0.28 to 0.7] |
| Polarity of reviewers' comments | 0.166 | 0.158 | 0.156 | 0.161 | 0.002 |
| | (0.189, 1.1806) | (0.189, 1.1717) | (0.189, 1.1688) | (0.189, 1.1747) | (0.222, 1.0020) |
| | [−0.2 to 0.54] | [−0.21 to 0.53] | [−0.21 to 0.53] | [−0.21 to 0.53] | [−0.43 to 0.44] |
| Number of citations | | | | 0.001* | 0.001* |
| | | | | (0.001, 1.0010) | (0.001, 1.0010) |
| | | | | [−0.0 to 0.0] | [−0.0 to 0.0] |
| Variance of reviewer comments' polarity | | | | | 0.300 |
| | | | | | (0.220, 1.3499) |
| | | | | | [−0.13 to 0.73] |
| Observations | 4,845 | 4,845 | 4,845 | 4,845 | 4,845 |
| Log Likelihood | −18,146.530 | −18,145.360 | −18,145.030 | −18,143.640 | −18,142.720 |

Note:
Standard error and odds ratio are in parentheses: (std, odds); 95%CIs are reported in brackets: [lower, upper] $^*p < 0.1$; $^{**}p < 0.05$; $^{***}p < 0.01$.

(including reviewer response) and the submitted manuscript's contribution potential. We then study the reviewers' comments. More specifically, we aim to understand which author or reviewer characteristics are associated with peer review sentiment. Finally, we focus on the modifications authors make to their manuscripts during the peer review process, classifying the words added to the manuscript during the first-round of revisions.

## Prediction of a paper's acceptance timeline

In this analysis, we focus on the revision decision during the first review iteration. Table 3 shows the results of the conditional logit models on the editor's revision decision.

We report the distribution of the number of revisions prior to acceptance in Table A2 in the Appendix.

We use five different models reflecting different variable permutations, the results of which are represented by the five columns in Table 3, where column 1 corresponds to model 1, column 2 corresponds to model 2 and so on. For models 1 and 2, we try different h-index measurements. In model 1, we use the maximum h-index out of all contributing authors of an article, and use the average h-index of the contributing authors for model 2. Between models 1 and 2, we see that the average h-index (model 2) has a higher correlation with the editor's decision to not ask for a subsequent revision. We incorporate authors' gender information in models 3 and 4, and in model 4, we use the number of citations to control for the paper's contribution potential. Finally, model 5 incorporates the variance of the polarity of the individual reviewer's comments that we use as a proxy measurement for the degree of controversy of the review for the particular paper.

In all models, we observed that the h-index of the authors decreases the probability of receiving a request for additional revision(s). This finding indicates that authors with higher h-indexes may be able to produce papers that are closer to publication quality, possibly due to more experience in publication. We also observed that gender does not affect the revision decision process. These results do not quantitatively change across all of the models in Table 3. These results also imply that the authors' h-index metrics increases the acceptance speed (fewer rounds of revisions before the editor accepts the paper) regardless of the paper's validity as assessed by the reviewers.

We also conducted analysis to test the possible biases in the inclusion of an author's h-index. As discussed in the Methods section, we use the authors' "current" h-index. Therefore, our h-index data is a snapshot at the time of data collection (October 28, 2019) and not the h-index at the time of each paper's submission. We acknowledge this limitation of our study, as the h-index history for each individual author is not attainable from any open data source that we are aware of. To estimate the possible effects of any temporal gap, we ran the logit-regression model with the data that was published within one year after the h-index was retrieved (*e.g.*, papers in our dataset that were published from October 28, 2018 through October 28, 2019) in Table A5. We found that the coefficients are almost identical and are statistically significant. We then conduct the same analysis as Table 3 using a different NLP sentiment analysis library, Vader (*Bird, Klein & Loper, 2009*) to replicate the findings (see results in Table A4). Using a different sentiment analysis tool yields similar results, and the results from these models support our previous findings.

## Prediction of contribution potential

Table 4 shows the results of our linear mixed model regressions. We focus on publications that were published before 2019, which have had more than 10 months to accrue any citations—using these papers does not change our findings in any appreciable way. There were 3,945 articles that were published prior to 2019 (Table A1). Model 1 considers both fixed and mixed effects, while model 2 only takes into account fixed effects, and we
**Table 4 Linear mixed model: the number of citations.**

| | The number of citations | | The number of citations (log) | |
|---|---|---|---|---|
| | (Model 1) | (Model 2) | (Model 3) | (Model 4) |
| Number of revisions | −1.904 | −1.152 | −0.049 | −0.068** |
| | (2.113) | (0.740) | (0.089) | (0.032) |
| Number of authors | 0.849** | 0.741*** | 0.069*** | 0.070*** |
| | (0.392) | (0.168) | (0.015) | (0.007) |
| Year-month trend | 0.021 | 0.021*** | 0.003* | 0.003*** |
| | (0.880) | (0.002) | (0.039) | (0.000) |
| Average h-index | 0.396 | 0.316*** | 0.026*** | 0.026*** |
| | (0.173) | (0.056) | (0.005) | (0.002) |
| Research category dummy | 0.338 | −0.465 | −0.015 | −0.023*** |
| | (0.411) | (0.322) | (0.043) | (0.008) |
| Intercept | −756.543 | 1019.768 | 52.603 | 49.309*** |
| | (909.132) | (711.238) | (73.550) | (16.738) |
| Observations | 3,945 | 3,945 | 3,945 | 3,945 |
| Log Likelihood | −19,176.6112 | −19,116.7643 | −6,815.9456 | −6,710.6895 |

Note:
Standard error in parentheses. $*p < 0.1$; $**p < 0.05$; $***p < 0.01$. The table shows only the fixed effects for each model. (1, 3): fixed and mixed effects, (2, 4): only fixed effect. Year-month trend is a linear trend to consider. Publication dates differences.

observe a negative fixed effect on the number of revisions. In both models 1 and 2, we observe that the coefficient of the number of revisions has a higher standard error than the coefficients of all other independent variables have. This high standard error implies that the number of revisions is not strongly related to the number of citations a publication receives. We conduct the same analysis on the logarithm of the number of citations (models 3 and 4), but the results do not change these implications.

As we did in "Prediction of a Paper's Acceptance Timeline", we use Altmetrics scores and readership numbers as alternative variables that proxy the performance of the paper in Table A7, and yield similar results.

## Review content sentiment

Table 5 shows the results of the linear mixed effect model on the reviewers' sentiment. In this analysis, we use 11,720 reviewers' sentiment in 4,871 articles (Table A1). For both subjectivity and polarity sentiments, we find that the reviews written by reviewers who opt to "sign" their reviews are more subjective and more positive than the anonymous reviewers' reviews. We did not observe a strong gender impact on either of the analyzed sentiments. We observe a negative coefficient for the ratio of female authors, which implies that authors who "sign" their reviews tend to give less subjective reviews for papers written by authors with a higher female author ratio. However, the standard error of the coefficient is still high and we conclude that this result might not be robust. Sentiment analysis using Vader (see results in Table A6[2]) generated the same findings as

[2] We only provide the analysis with the polarity score as NLTK does not calculate the subjectivity score.

**Table 5 Linear mixed model: Sentiment of reviewers' comments.**

| | Reviewers' sentiment | |
| --- | --- | --- |
| | (Subjectivity) | (Polarity) |
| Female first author | 0.001 | 0.002 |
| | (0.005) | (0.003) |
| Female author ratio | 0.016* | −0.004 |
| | (0.009) | (0.005) |
| Female author ratio * Name revealed reviewer | −0.020* | 0.009 |
| | (0.012) | (0.008) |
| Average h-index | 0.000* | −0.000 |
| | (0.000) | (0.000) |
| Name revealed reviewer | 0.010** | 0.013*** |
| | (0.004) | (0.003) |
| Article dummy | −0.000 | −0.000 |
| | (.000) | (0.000) |
| Intercept | 0.436*** | 0.121*** |
| | (0.005) | (0.003) |
| Observations | 16,933 | 16,933 |
| Log Likelihood | 8,717.6418 | 12,136.3514 |

Note:
Standard error in parentheses. $*p < 0.1$; $**p < 0.05$; $***p < 0.01$. This table shows only each model's fixed effects.

TextBlob (see results in "Prediction of a Paper's Acceptance Timeline"), further validating the implications of our findings.

## Taxonomy of manuscript modifications during a revision

In our LDA model, we set the number of topics to 13[3] and train the model on the documents from *PeerJ—Journal of Life and Environmental Sciences*[4] (a total of 4,899 documents with an average of 580 words per document). Figure 2 shows the top 10 most frequent words in each topic. We color code the words to differentiate the different perceived connotations of a particular group of words. We note that this categorization of words is our speculation on what these most frequent words might be insinuating about the words that have been added to manuscripts during the first revision round. The blue words represent words describing the results (*e.g.* study, analysis, view). We color the words related to a paper's evidence with green (*e.g.* data, table, figure). For example, when data is added to the updated manuscripts, the authors might add additional evidence or description to their analysis. The orange words represent words that indicate a change in perspective (*e.g.* however, may, also). Because we only train our model on the words that have been added in the first revision, we interpret the words in orange as words authors use to change the implications in their papers. For example, the word "may" can change the insinuations of arguments in the authors' original submission. We leave research topic related words in black.

Figure 2 clearly shows that the majority of the top words are words related to analyzing the results (blue). This implies that authors add additional analysis or descriptions of their

[3] Decided by the perplexity and coherence scores described in Fig. A1 in the Appendix.

[4] These documents are the added words as described in "Manuscript Modifications".
| | 1 | 2 | 3 | 4 | 5 | 6 | 7 | 8 | 9 | 10 |
|---|---|---|---|---|---|---|---|---|---|---|
| Topic 1 | study | participants | data | table | results | time | research | health | control | journal |
| Topic 2 | data | study | model | population | species | fig | analysis | may | one | using |
| Topic 3 | patients | study | group | data | however | used | also | model | number | may |
| Topic 4 | data | textile | species | also | using | fig | used | two | journal | pmo |
| Topic 5 | data | study | used | analysis | page | figure | two | different | effect | one |
| Topic 6 | species | data | prey | table | using | also | figure | one | however | fig |
| Topic 7 | species | coral | water | study | marine | data | fish | reef | page | table |
| Topic 8 | soil | fig | water | carbon | using | plant | analysis | data | also | used |
| Topic 9 | fig | view | figure | dorsal | species | posterior | new | anterior | lateral | also |
| Topic 10 | species | data | genes | table | gene | using | genome | sequences | analysis | figure |
| Topic 11 | species | figure | using | different | time | also | females | data | used | may |
| Topic 12 | species | soil | ecology | plant | data | study | table | two | diversity | using |
| Topic 13 | cells | expression | cell | genes | protein | using | gene | study | fig | used |

**Figure 2** **The top 10 words in each topic estimated by Latent Dirichlet Allocation (LDA).** The colors represent the aim of the words (blue:analysis, green:evidence, orange:change in implication or perspective).

results in the first revision. Peer reviewers typically leave feedback on the results and findings of a paper. The aforementioned implication corroborates the view that authors are responding to and revising their manuscripts to encompass this feedback. Following this line of interpretation, it is easy to contextualize why evidence related words (green) are the second most frequent "type" of word to appear in the top words table. We also see that, in some revisions, authors add uncertainty (orange) to their revised manuscripts.

Namely, the results of our study suggest that the peer review process does indeed ensure that manuscripts that are eventually accepted meet a certain publication quality threshold, as there was no correlation between contribution potential and the number of review rounds a manuscript goes through. While we do find that some external factors, such as the authors' h-indexes, are associated with the editor's final decision of whether to request an additional revision or accept the paper, author gender and author h-index do not impact the reviewer's feedback sentiments. Finally, we find that the social pressures from a reviewer revealing their identity influences the sentiment polarity of that reviewer's feedback.

## DISCUSSION

This study first investigated how the revision process influences a manuscript's acceptance timeline and contribution potential. In our dataset, the editor is tasked with considering the peer reviewers' opinions and his or her own beliefs when deciding whether or not to ask the authors for an additional revision of the manuscript or to accept it to be published[5]. We evaluated factors related to the manuscript itself and external social factors that may impact the editor's verdict. As *Buljan et al. (2020)* posit, the influences of a paper's external social factors on the editor's verdict can be indicators of biases in the peer review process. However, we also recognize the possible confounding

[5] As our dataset does not include rejected papers, we are unable to observe the review history for a manuscript that is rejected. Thus editors, in our dataset, are part of the review process that eventually leads to the manuscript being accepted and published.

associations between a paper's external social factors and its own quality. For example, a well-received manuscript may be linked to both good paper quality and more experienced authors; the fact that the manuscript is from a group of experienced authors should not be overly interpreted and is not sufficient to conclude a bias on the part of the reviewer or editor. With this premise in mind, we find that the authors' reputation in academia plays a role in the eventual revision decisions. We also find that the number of revisions requested over the peer-reviewing process does not have a correlation with the contribution potential of the paper. We interpret this lack of correlation as confirmation that the review process, not the number of revisions an author has to make, ensures that published papers meet a certain quality criteria prior to publication. We then wanted to determine whether author or reviewer attributes are correlated with the review sentiment. We find that authors' gender and reputation do not affect the sentiment of the reviewer's feedback. However, given that reviewers are able to "sign" their reviews after a paper has been published, we find that reviewers who choose to identify themselves by signing their reviews write more subjective and positive reviews. This implies that social pressure from name association can influence the sentiment and polarity of a peer reviewer's publicly available review. Finally, we studied the modifications authors made to their papers in response to reviewer comments. During the manuscript revision process, we observe that, after receiving the peer reviewers' feedback, authors tend to add words that are related to analysis and that places bounds on their original arguments and results.

Our paper contributes to the current literature by deconstructing and scrutinizing the peer-review process itself. We studied the role of external factors such as gender in the context of the peer-review process (*Grogan, 2019*; *Murray et al., 2019*; *Laycock & Bailey, 2019*). While there is a controversy over whether gender biases exist and affect the peer review system and its results, our study found that authors' gender-related factors do not significantly predict the acceptance timeline or the reviews' feedback sentiment in this particular dataset. This finding is consistent with a previous study that focuses on the linguistic characteristics in peer review processes (*Buljan et al., 2020*). We also studied the modifications that authors made during the revision process. Whereas most literature on peer review concentrates on reviewer-side variables such as review sentiment or a reviewer's suggestion against or for acceptance, literature studying the peer review process from the author's viewpoint, such as manuscript revision based on reviewer feedback, is still lacking. Studying this unexplored side of the process is essential in our pursuit to understand how the peer review system maintains the quality of published scientific papers (*Bornmann, 2011*) and its credence (*Hemlin & Rasmussen, 2006*).

## CONCLUSIONS

Peer review is an integral part of the publication process that allows peers to review and discuss improvements to a paper in order to prepare it for publication in a journal. We focus on the peer reviewers' feedback and the subsequent discussion between the authors and peer reviewers through the rebuttals, manuscript revisions and follow up feedback.

We approach the analysis of the peer reviewing process with an emphasis on the content of the individual reviews and implications of the manuscript modifications. Given that our dataset does not contain the review histories of any rejected papers, the main function of the reviewers in our dataset is to correspond with the authors to improve the authors' paper to the point where the manuscript can be accepted for publication. Although revision decisions and sentiment of the peer reviewers' feedback may vary, our study implies that the process of peer-reviewing plays a role in ensuring a accepted and published paper meets a certain baseline of quality and contribution potential.

There are still many questions that we were unable to address within this study. Our analysis is limited by the lack of review audit history from papers that were either rejected or eventually rejected by the editor. This missing dataset is critical to the robustness of our analysis, and we hope that, in future extensions of this paper, we are able to obtain this additional data. Another limitation of our study is the breadth of research areas that the manuscripts in our dataset covers, as our publications are currently only from natural science and computer science journals and are all published in PeerJ journals. In addition, our taxonomy study of manuscript modification only describes the categories that certain words are generally associated with. However, in our current study, we were unable to measure how exactly added words changed a specific paper's implications during the revision process. For example, we found that words such as "however" and "may" were added during a revision. While we categorized these as words that are generally associated with a change in implication or perspective, it is not explicitly clear whether such words weakened or changed the statements that an author argued in a specific manuscript.

In the future, we hope to incorporate review audit histories from other research areas (such as the humanities) and from other publishers. Finally, our study focuses only on the additions authors make to their manuscripts during the first round of revision, and we plan to expand this to encapsulate both the content removed in a revised manuscript and subsequent revisions made in the entire review history.

# APPENDIX

## Descriptive statistics

**Table A1 Number of articles and reviewers.**

| | Size |
|---|---|
| # of Articles for Contribution Potential Analysis (as of 2018 year) | 3,945 |
| # of Articles for Sentiment Analysis | 4,871 |
| # of Reviewers for Sentiment Analysis | 11,720 |

**Table A2 Distribution of accepted revision.**

| Accepted Revision | # Papers |
|---|---|
| After 1st revision | 2,714 |
| After 2nd revision | 1,745 |
| After 3rd revision | 315 |
| After 4th revision | 57 |
| After 5th revision | 9 |
| After 6th revision | 5 |

**Table A3 Summary of variables.**

| Variable | Description |
|---|---|
| Maximum h-index | The maximum h-index among the authors. |
| Average h-index | The average h-index of the authors. |
| Female last author | Dummy variable that is 1 when the last author is female, otherwise is 0. |
| Female 1st author | Dummy variable that is 1 when the first author is female, otherwise is 0. |
| Female author ratio | The ratio of female of the authors. |
| Polarity of authors' rebuttal | The polarity of the authors' rebuttal. |
| Polarity of reviewers' comments | The polarity of the reviewers' comments |
| Variance of polarity of reviewers' comment | The variance of the polarity of the reviewers' comments. |
| Number of citations | The number of citations of the article. |
| Number of revisions | The number of rounds of revision that the article underwent. |
| Number of authors | Number of authors of the article. |
| Year-month trend | Linear trend variable for the time elapsed since a paper's publication. |
| Research category dummy | Dummy variable for research category. |
| Article dummy | Dummy variable for articles. |
| Name revealed reviewer | Dummy variable that is 1 if the reviewer reveals his or her name, otherwise is 0. |

# Prediction of the paper acceptance timeline

**Table A4 Conditional logit models: revision decision (NLTK sentiment calculation).**

| | Accepted at the first revision | | | | |
| --- | --- | --- | --- | --- | --- |
| | (Model 1) | (Model 2) | (Model 3) | (Model 4) | (Model 5) |
| Maximum h-index | 0.005*** | | | | |
| | (0.001) | | | | |
| Average h-index | | 0.012*** | 0.012*** | 0.012*** | 0.012*** |
| | | (0.002) | (0.002) | (0.002) | (0.002) |
| Female last author | | | −0.044 | −0.043 | −0.044 |
| | | | (0.057) | (0.057) | (0.057) |
| Female 1st author | | | −0.003 | −0.0004 | −0.001 |
| | | | (0.053) | (0.053) | (0.053) |
| Female author ratio | | | 0.036 | 0.034 | 0.042 |
| | | | (0.114) | (0.114) | (0.114) |
| Polarity of authors' rebuttal | −0.026 | −0.021 | −0.021 | −0.020 | −0.022 |
| | (0.037) | (0.037) | (0.037) | (0.037) | (0.037) |
| Polarity of reviewers' comment | 0.039 | 0.039 | 0.039 | 0.040 | 0.030 |
| | (0.060) | (0.060) | (0.060) | (0.060) | (0.060) |
| Number of citations | | | | 0.001* | 0.001* |
| | | | | (0.001) | (0.001) |
| Variance of polarity of reviwers' comment | | | | | −0.199*** |
| | | | | | (0.075) |
| Observations | 4,845 | 4,845 | 4,845 | 4,845 | 4,845 |
| Log Likelihood | −18,146.710 | −18,145.660 | −18,145.350 | −18,143.990 | −18,140.410 |

**Note:**
Standard error in parentheses,*$p < 0.1$;**$p < 0.05$;***$p < 0.01$. The sentiment is calculated by Vader in the NLTK library (*Bird, Klein & Loper, 2009*).

**Table A5 Conditional logit models: revision decisions for manuscripts published between October 28, 2018–October 28, 2019.**

| | Dependent variable: | | | | |
|---|---|---|---|---|---|
| | Accept | | | | |
| | (1) | (2) | (3) | (4) | (5) |
| Maximum h-index | 0.006** | | | | |
| | (0.002) | | | | |
| Average h-index | | 0.011** | 0.011** | 0.011** | 0.011** |
| | | (0.005) | (0.005) | (0.005) | (0.005) |
| Female last author | | | −0.135 | −0.141 | −0.142 |
| | | | (0.125) | (0.126) | (0.125) |
| Female 1st author | | | −0.039 | −0.046 | −0.048 |
| | | | (0.118) | (0.118) | (0.119) |
| Female author ratio | | | 0.077 | 0.087 | 0.087 |
| | | | (0.257) | (0.258) | (0.258) |
| Polarity of authors' rebuttal | 0.674 | 0.646 | 0.653 | 0.681 | 0.670 |
| | (0.685) | (0.685) | (0.684) | (0.686) | (0.686) |
| Polarity of reviewers' comments | −0.226 | −0.185 | −0.175 | −0.177 | −0.046 |
| | (0.462) | (0.461) | (0.461) | (0.461) | (0.532) |
| Number of citations | | | | −0.017 | −0.018 |
| | | | | (0.030) | (0.030) |
| Variance of polarity of reviwers' | | | | | −0.270 |
| | | | | | (0.549) |
| Observations | 1,102 | 1,102 | 1,102 | 1,102 | 1,102 |
| $R^2$ | 0.006 | 0.004 | 0.005 | 0.006 | 0.006 |
| Log Likelihood | −2,648.903 | −2,649.939 | −2,649.237 | −2,649.065 | −2,648.943 |

Note:
  Standard error in parentheses, $^*p < 0.1$; $^{**}p < 0.05$; $^{***}p < 0.01$.

## Sentiment analysis

**Table A6 Linear mixed model: Sentiment of reviewers' comments (NLTK Vader sentiment calculation).**

|  | Reviewers' sentiment (Polarity) |
| --- | --- |
| Female 1st author | −0.006 |
|  | (0.018) |
| Female author ratio | −0.049 |
|  | (0.036) |
| Female author ratio * Name revealed reviewer | 0.052 |
|  | (0.046) |
| Average h-index | 0.000 |
|  | (0.000) |
| Name revealed reviewer | 0.056*** |
|  | (0.017) |
| Article dummy | −0.000 |
|  | (0.000) |
| Intercept | 0.621*** |
|  | (0.020) |
| Observations | 16933 |
| Log Likelihood | −14089.1895 |

**Note:**
Standard error in parentheses, *$p < 0.1$; **$p < 0.05$; ***$p < 0.01$; Only the fixed effects of each model are shown.
The sentiment is calculated by Vader from the NLTK library *Bird, Klein & Loper (2009)*.

**Table A7 Linear mixed model: Altmetrics and # readers**

|  | *Altmetrics* | | *Readers* | |
| --- | --- | --- | --- | --- |
|  | **(Model 1)** | **(Model 2)** | **(Model 3)** | **(Model 4)** |
| Number of revisions | 0.868 | 6.300 | −0.303 | 5.285 |
|  | (2.185) | (7.455) | (1.949) | (6.385) |
| Number of authors | 1.512*** | 3.207 | 2.545*** | 5.753* |
|  | (0.494) | (2.154) | (0.440) | (3.052) |
| Year-month trend | 0.009 | −0.013 | 0.043*** | 0.045 |
| Average h-index | 0.486*** | 0.972*** | 0.986*** | 0.416 |
| Research category dummy | 2.818 | 4.965 | −0.781 | 2.235 |
|  | (1.219) | (5.976) | (1.198) | (3.420) |
| Intercept | -6,209.696** | −11,294.495 | 1,723.090 | −4,961.586 |
|  | (909.132) | (711.238) | (2,646.4580) | (7,554.786) |
| Observations | 3,888 | 3,888 | 3,888 | 3,888 |
| Log Likelihood | −23,009.8109 | −23,109.4955 | −22,568.5619 | −22,585.5649 |

**Note:**
Standard error in parentheses, *$p < 0.1$; **$p < 0.05$; ***$p < 0.01$; The fixed effects on the each models are only shown; Model 1, 3: only fixed effects, Model 2, 4: fixed and mixed effects.

## Selection of the number of topics for LDA topic modeling

We utilize the perplexity and coherence scores to determine an appropriate number of topics for LDA to use as a hyperparameter. While the perplexity is one of the most popular metrics for selecting the number of topics *Wallach et al. (2009)*, it does not always return the optimal number of topics (*Chang et al., 2009*). To conduct a robust analysis, in addition to the perplexity, we also used the coherence scores (*Stevens et al., 2012*). By calculating those two metrics for different numbers of topics, we try to use the number of topics where the perplexity is low and the coherence score is high.

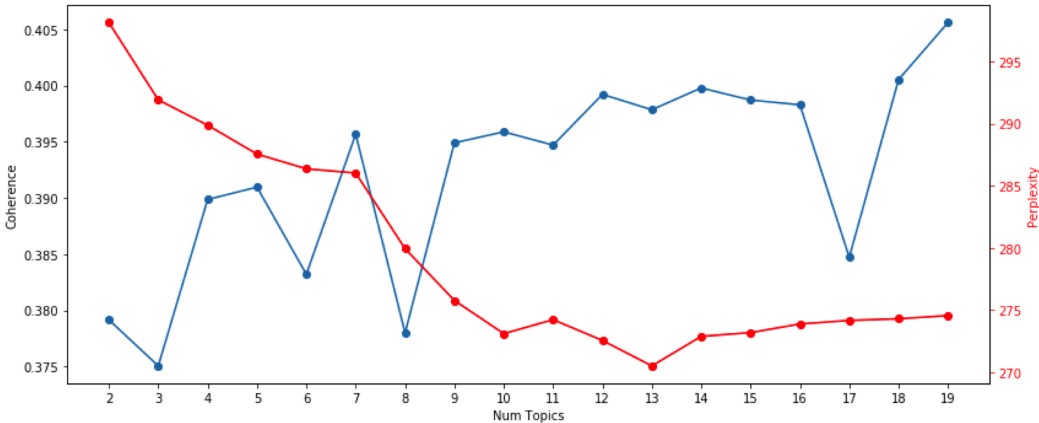

**Figure A1** **The perplexity and coherence score.** The perplexity and coherence score for different # Topics for Latent Dirichlet Allocation (LDA) (Perplexity: red, right y-axis; Coherence score: blue, left y-axis).            

## Funding

This work was supported by DARPA (grant no. D16AP00115 and contract no. W911NF-17-C-0094). The funders had no role in study design, data collection and analysis, decision to publish, or preparation of the manuscript.

## Grant Disclosures

The following grant information was disclosed by the authors:
DARPA: D16AP00115 and W911NF-17-C-0094.

## Competing Interests

The authors declare no conflicts of interest.

## Author Contributions

- Akira Matsui conceived and designed the experiments, performed the experiments, analyzed the data, prepared figures and/or tables, authored or reviewed drafts of the paper, and approved the final draft.

- Emily Chen conceived and designed the experiments, prepared figures and/or tables, authored or reviewed drafts of the paper, and approved the final draft.
- Yunwen Wang conceived and designed the experiments, authored or reviewed drafts of the paper, and approved the final draft.
- Emilio Ferrara conceived and designed the experiments, authored or reviewed drafts of the paper, and approved the final draft.

## Data Availability

The code is available at Zenodo: https://zenodo.org/record/5199504.

This paper was written using data obtained on December 1, 2019 from Digital Science's Dimensions platform, available at https://app.dimensions.ai. The details and examples of queries for retrieving publication information can be found in the official documentation at https://docs.dimensions.ai/dsl/. Access was granted to subscription-only data sources under license agreement. Digital Science may provide access under their scientometric data access program. PeerJ articles and reviews were retrieved from their online archive, which are all published under Creative Commons Attribution license (CC-BY).

## Supplemental Information

Supplemental information for this article can be found online at http://dx.doi.org/10.7717/peerj.11999#supplemental-information.

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
