# Peer review of "The impact of peer review on the contribution potential of scientific papers"

_PeerJ, doi:10.7717/peerj.11999_

## Round 0.1 · original submission · Major Revisions

Two of the reviewers agree that the paper is very interesting, however, there are some common themes in their criticisms that you will need to pay particular attention to:

1. The research question of the study is not clearly defined.
2. More relevant literature should be reviewed.
3. The question of why papers that were reviewed but ultimately rejected were excluded from your study and how that affects the results.

Please also provide a detailed response to each of the reviewers' comments with your revised paper.

·

Basic reporting

The paper is clear but the authors should explicitly mention, from the abstract and the introduction, that the work is only dealing with manuscripts accepted and published.

Raw data for LDA preprocessing and sentiment calculation is not available.

Experimental design

I am concerned about which h-index is used for the analysis: the "current" one or the one the authors had when the paper was first submitted. From your interpretation in Section 5.1, I guess the latter would be the right one but, if Dimension API does not provide it, you would need to calculate it from the citations received before the paper submission.

Sentiment is calculated using TextBlob's default configuration (i.e. NLTK Naive Bayes classifier on the whole text). The authors could check that the results hold for other more sophisticated approaches (e.g. weighting positive/neutral/negative sentences using SentimentR or following a neural network based approach as in Stanford CoreNLP).

More information is needed on how added words are collected for measuring manuscript modification. How can this process be affected by the fact that new words may appear as a result of text changes or deletions? How does this process deal with word changes such as making a term plural, using different variations of the same root (e.g. see fig and figure in Topic 9 of Figure 2) or having different hyphenation from the PDF? What text cleaning and preprocessing has been applied after text extraction?

Two unbalanced research areas are considered in this study (i.e. 4917 vs 165 articles in Table 1). I wonder how this can affect the models in Tables 1 and 4.

Validity of the findings

The models in Table 2 should also look at the number of reviewers reviewing an article and at the level of agreement of their reports or recommendations, since the editorial decision may be also driven by these variables. The relationship between the polarity of authors and reviewers might be further analyzed to better understand the dialogue established during the peer review process.

The selection of the number of topics (i.e. 13) in Figure 2 should be further justified, as well as the effect of the threshold set for the minimum frequency (i.e. 50). The manual grouping of words should be explained into more detail and it should be numerically analyzed, maybe beyond the Top 10.

Table 3 should look at the reviewer recommendation as previous research has found the relationship between the sentiments of reviewing reports and the reviewer recommendation (e.g. https://elifesciences.org/articles/53249, do not need to cite, I am sorry for the self-reference). Indeed, previous research has found that reviewers providing a positive recommendation (mainly "accept" or "minor revisions") are those more willing to sing their reviews, which would also be in line with your findings of more subjective and positive reports.

Table 4 concentrates on the number of citations but other Altmetrics could have also be considered (e.g. number of reads).

Additional comments

Congratulations for this very interesting and nice work.

·

Basic reporting

Needs to be improved to be more reader friendly. Literature needs to be expanded.

Experimental design

Research questions are defined 3 times differently, a clear list of outcomes and exploratory variables is needed.

Validity of the findings

Until authors address my questions, I am unsure of the validity of the findings.

Additional comments

Thank you for the opportunity to review this manuscript. I would like to offer my suggestions for its improvement:
Introduction
1. Reviewers are also able to leave feedback, which enables the authors of the submitted paper to revise the paper to improve upon or extend the paper. – this is redundant, u mention that with minor and major revision
2. to improve the submitted work – you failed to mention here the role of peer review in improving the reporting not the work itself, those two aspects should be clearly differentiated
3. You goal says that you look at contribution potential, yet this is not described in the abstract. And in the methods and results, you mix this with number of revisions. According to your definition this should be citations of a paper or Altemtric scores. Aditionally, if a journal has like peer j, has an impact factor, and this represents the average number of citations of paper for that journal – how likely is it that you would find any correlations with the things you investigate when most papers should fall closely around that citation number.

Background

1. The first paragraph of your background says nothing on the contribution potential as you define it.
2. we wonder to what extent the increased efficiency translates to the equability and quality of a paper’s eventual contribution potential. – this does not belong to lit. review section
3. You should include the following study and its findings in your lit. review – missing this, it is not clear how your lit. search was conducted: 10.1186/s12874-019-0688-x
4. we propose to explore a use of objective measurement for accessing peer review quality and their impact on paper contribution potentials. – If this was your goal it should not be under lit. review section, also it is not mentioned in the abstract, nor do I see it anywhere in results.


PeerJData

1. typically invited by the editors of the journal – please elaborate, how many are not invited by editors? How many of those are in your sample?
2. Please specify if reviewers make recommendations (minor, major, rejection)
3. Table 1, please include data on number of all articles, and then percentage of those with open reviews. Additionally, what if one review is open and another is not for the same paper – how is this counted?


The Impact of Revisions
1. We show that the number of revisions does not correlate to the paper’s contribution potential,
we note here that as we were unable to obtain a data set that includes the revision history of rejected papers, our conclusions only hold true for accepted and published manuscripts. – this is results, should not be in methods.
2. If you are correlating number of revisions and citations why not state so in abstract and methods, instead of saying contribution potential – the former is clearer for the readers.

Results

1. Please start results with a clear description of the number of analyzed manuscripts, how many reviewers they had, and how many revisions, what were the most common decision after round 1 stage, and how many had more than 2 review rounds.
2. Table 2, please provide Odds ratios
3. In all models, we observed that the h-index of the authors decrease the probability of receiving request for additional revision(s). Please quantify this and be clear do you mean after review 1 round – how many papers were accept in initial stage?
4. status in the academia – how is this defined, and how different is it from h index?
5. Please provide in the methods table of all collected variables, and specify why did you not control for all of them in your models? You seem to have review decision, field, number of reviewers, sentiment, number of revisions, gender of authors (female ratio, female last authors, female first author – and these are very likely intercorrelated – how do you account for that), polarity of comments and rebuttals, citations and so on – so please list all in a table, and indicate how they were included in each model.
6. How many of the manuscripts had the same authors? Or overlapping authors?


Topic modeling
1. this analysis insinuates that during the manuscript modification process, authors add additional analysis
313 and evidence and may alter the implications of the results from their original papers. – I don’t think you have proof of this, please take 10% of the sample and manually compare have there actually been changes in analysis – also clearly define what you mean by change in analysis ? for example adding OR is that a change in analysis or change in reporting? Increasing or decreasing decimal points? Moving things from or to supplement?

2. Figure 2 clearly shows that the majority of the top words are words related to analyzing the results – nothing clear there when I look at it – most common blue word is study – it could be found in intro, in discussion, in abstract – what has that word have to do with analysis – how do you know it is used as a verb and not a noun?

Overall, I find the results poorly written as there is too much talk of models, and less on findings. Please make the language understandable to readers – e.g. please start the results (after descriptors I asked above) with: we found that h-index of the authors decrease the probability (OR XY) of receiving a requests for additional revision(s) (finding confirmed by 4 models, see Supplementary fie) and remove unnecessary text of modeling, do the same for all other findings. In the supplement, please provide n for the models, and it isn’t clear how many manuscripts with more than 1 revision you had. Additionally, its seems you only looked at acceptance after first revision, what about characteristics after 2nd or 3rd revision, were all those lumped together as control?


Discussion

1.Please start first with your main findings. In our study of xy PEER J manuscirpts, we found that….
2. We find that the number of revisions requested over the peer-reviewing process does not have a
346 correlation with the contribution potential of the paper. – it isn’t clear here what is the contribution potential of the paper, as in methods you said it’s the number of revisions?


Final comments:

1. It is not clear what relations authors have to PeerJ, and what is PeerJ policy on data scraping. Was PeerJ contacted to provide info or participate in the study?
2. Please use STROBE to report the study and make it clearer.


In hopes may comments can help you improve your manuscript,
Kind regards,
Mario Malicki

Reviewer 3 ·

Basic reporting

The authors report about a study on open reviews of the journal PeerJ. The go into the details of the written reviews. They analyse communications between reviewer committees and authors regarding the revisions on specific paper.

The paper is well-written.
For a journal paper, the amount of references seems not too comprehensive. Introduction and related work are easy to read but could be extended.
The structure of the paper follows the typical format.

Experimental design

The authors write: "our study provides insight into the content of peer reviews and the subsequent modifications ..." ... "Therefore, we propose to explore a use of objective measurement for accessing peer review quality and their impact on paper contribution potentials" This claim holds true: They provides some insights, ... they explore ... but they do not provide clear evidence of a critical process during peer review.

A concrete RQ with an clear answer is missing. This is the major weakness of this paper.

In general their experimental design is sound.

The methods and techniques used in the study make sense but I miss a concrete discussion why the authors used this and that tool. E.g. they used TextBlob for the measurement of polarity of texts. Why this tool? Why LDA in the described configuration (e.g. Words that appear less than fifty times)? We lean not enough on this in the paper.

I am not sure why the authors excluded the rejected papers. Sure, desk rejected papers can be excluded, but paper which got 1 or more revision and are then reject could have been a nice control sample. Authors could report about this.

Validity of the findings

The regression analysis to generate the results looks correct.

E.g. they say "In all models, we observed that the h-index of the authors decrease the probability of receiving a request for additional revision(s)."

This result sounds logic but is not really discussed. Of course, experienced authors with a reasonable h-index, produce papers that a readable and structured in the understandable manner.

I am also not too happy with the inclusion of bibliometric metrics into the analysis. High h-index of authors can be a side indication but should not over interpret.
I suggest to perform more analysis on the validity of the reviewer comments and the straightness of the revision and readability of the rebuttals.

Additional comments

In general, a very nice paper with a good contribution.
See 3 for the weaknesses.

As this is early reseach, I miss the main 3+ take aways and reasonable next steps.

The paper seems like a short paper for me. Most parts could be extended.
In my view, the paper takes not the most logic approach and slips away with the inclusion of bibliometrics metrics and sentiments.

"We use the h-index of the author as a proxy to quantify his or her prestige or reputation in academia" ok, but why? I do not believe that reviewers check the h-index of an author before starting the review. Or do they? If yes, please provide evidence that this is standard.

Including these metrics is too early. Too many things are unclear and undisputed, to start with indicators here.

Again, the topic and the dataset of the paper and study seem to be very precious and sound. But explain better why you use which technique and boil it down to the most fundamental questions and answer.

I like to review a 2nd version of the paper.

---

## Round 0.2 · Minor Revisions

One of the referees recommended to accept your paper, however, the two others have a long list of issues, which need to be resolved prior to publication.

·

Basic reporting

I found tables in the Appendix difficult to follow because they are not uniquely identified (eg. three different Table A1).

Table A2 (the alternative to Table 3) is not analyzed within the text.

Can you please report on the time window covered by the PeerJ Data? Collection date is specified but I am afraid I have missed the date of the earliest submission. If it was far from the collection date, it might also be far from the current h-index values and have an influence on the interpretation of the models in Table 1.

Please, double check for typos (eg. The results of our study __suggests__ ..., Number of __author__, polarity of __reviwers__, etc.).

Experimental design

When comparing the results for the h-index (Table 2) with those for the average number of affiliations (Table A2), I am worried about them being inversely correlated with the editors decision.

Also, when comparing Table 2 with Table A1, I am concerned about the different effect of the variance of polarity of reviewers’ comments. Can you further elaborate on this?

Does the variable "Number of revision" in Table 3 and Table A2 mean the number of review round (as defined in Table A3) or the number of reviewers involved? More reviewers are expected to provide more feedback, which might generate more manuscript modifications and affect the contribution potential.

Regarding text preprocessing, I wonder if you are running any kind of word stemming (e.g. see fig and figure in Topic 9 of Figure 2) and if that would have any effect on your results.

Validity of the findings

I wonder whether the interpretation "that the h-index of the authors decrease the probability of receiving a request for additional revision" (line 349) could be seen all the way around. Could the results be actually saying that authors receiving more positive editorial decisions end up having a higher h-index?

Do the models in Table 4 account for the reviewer recommendations? Do reviewers sign because they are actually recommending "accept or "minor revision" and, in turn, the type of recommendation is ruling the values of polarity and subjectivity?

Additional comments

The authors have mostly solved my previous concerns and I am happy with their responses.

Still, I have some minor comments shown above.

·

Basic reporting

Paper structure needs to be improved. Methods section and ordering must be the same as in the results, with all variables and their definitions clearly stated.

Experimental design

Most of the methods are clearly described, but variables need to be better defined. Additionally, authors continue to use the term contribution potential, when in fact they are only talking about papers citations - and in Table 2 they call it number of citations. There is no need to talk about contribution potential unless they specify how that differs from number of citations measured by Dimensions.

Validity of the findings

The authors do not report 95%CI, and it is my opinion that unless they manually compare differences between two versions for a sample of manuscripts and their revisions, they can not use the LAD method to claim that "authors add additional analysis and evidence" during revisions. LAD has not been validated as a method that can prove this in any previous studies.

Additional comments

Thank you for the opportunity to re-review this work. I find it improved, but I still find many issues that I suggest addressing:

Introduction
1) - Although the peer-reviewing process often feels like a heavy burden to many scientists, - citation is missing for this part of the sentence
2) During the review process, reviewers decide whether to accept, request a major or minor revision or reject a submitted paper. – this is not true for all journals, some do not ask reviewers for decisions
3) The results of our study suggest – this paragraph should not be in the introduction – leave results to results and discussion
4) influence the editor’s final decision – this study is observational, you are not looking at influence, but association


Literature review
1) Therefore, we aim to explore a use of
90 objective measurements for assessing peer review quality and their impact on paper contribution potential. – this is a different goal then the one you mentioned in the introduction

2) In our study, we operationalize “contribution potential” as the citation metrics and a paper’s
93 citation count. – what is the difference between metric and count? Finally, in intro u state: We define contribution potential as the 30 subsequent contribution and impact metrics that a paper amasses over time – Please have one clear definition of what contribution potential is.

Methods
All list of options for 4.1.3 and 4.1.4. should be listed (e.g. minor, major revisions, or type of changes for 4.1.4.)


Results
1) Table 2 – is missing 95%CI for Odds ratios, and with so many odds ratios being close to 1, the conclusions from these models are questionable.

2) Table 2 on the left column has a much clearer list of variables, than any you described in the methods section. I would strongly urge the authors, to have a list in the methods of variables, using the exact names as were used in Table 2 – and one sentence descriptions of their meaning. This would me much clearer for me as a reader to follow, then trying to decipher the meanings in the long paragraphs of current methods – and constantly going back to detect mismatches between what was written in methods, vs what is in the Table 2.

3) 5.1.2. We used publicly available Altmetric scores – this paragraph should be in methods not results

4) 5.3 My previous comment stands, authors should manually compare a sample of randomly chosen articles to demonstrate that authors add additional analysis and evidence. Without that, they must greatly tone down their results and can only claim indications of changes. Additionally, this section of results requires a statement on what percentage of manuscripts based on this method indicates additional analysis or evidence. Otherwise the usefulness of methods is very debatable.

Overall structure of results:
Please make the paper easier to follow, if your 4.1.1 is Peer Review Content, then results 5.1.1. should be about peer review content, not about acceptance timeline. Use the same numbering order as is presented in the results in the methods. In other words, if 4.2. is Peer review feedback, than 5.2. needs to have the same title.

Reviewer 3 ·

Basic reporting

The authors improved their paper largely.

Experimental design

The authors also improved the experimental design.

Validity of the findings

Findings look good!

Additional comments

Thank you for the successful revision.

---

## Round 0.3 · Major Revisions

One reviewer recommends to accept the paper and the other to reject. The latter is frustrated by your lack of response to his previous questions. Given this split advice, I will give you one more opportunity to revise the paper in line with the suggestions of the second reviewer. I won't send it back to him, but review it myself to see if you have successfully addressed his criticisms.

·

Basic reporting

The authors have very much improved the manuscript, which now looks complete and clear.

Experimental design

The experimental design is well defined and analyzed.

Validity of the findings

Findings are very interesting and valid. Some of them confirm previous work while other shed more light on the study of peer review.

Additional comments

Congrats, I like your work very much.

One final question, I wonder whether the conclusion " the process of peer-reviewing plays a role in a paper’s eventual contribution potential" is in line with the analysis in section 5.1.2 or with the sentence in the discussion referring to the "process does not have a correlation with the contribution potential of the paper".

·

Basic reporting

Biased interpretation of results.

Experimental design

Methods do not align with the statements made.

Validity of the findings

Conclusions overinterpret the data.

Additional comments

Thank you for the opportunity to again review this manuscript, but as my requests from the previous round were not met, I will only mention few things, and not provide a full review:
1) I may be now repeating myself for the 3rd time, but I find it strange that authors still do not clearly define some aspects of their study to the readers, e.g., why is there a need in the introduction to only partly define contribution potential. Specifically, the authors stated: (“e.g. the number of citations or altmetrics”), yet in the rebuttal letter they say that it is: “We used the number of citations, Altmetrics and the number of readers from reference manager software, such as Mendeley. More precisely, the paper’s citation count refers to the number of citations a paper receives and the citation metrics refers to Altmetrics and the readership numbers. We define the contribution potential of a given paper as how much recognition that article earned, which is calculated by both the citation metrics (readership counts) and a paper’s citation count.”
I have not once criticised your definition, only asked that it is clearly spelled out what it entails – so please do so for the readers. Comparatively, this is your definition in the literature section: “we operationalize “contribution potential” as the citation metrics and a paper’s citation count”; and in the methods section 3.3. you say: “Additionally, we used publicly available Altmetric scores and the number of readers from reference manager applications such as Mendeley to calculate the contribution potential of papers, which were retrieved through the Altmetric API.” (Why say such as Mendeley, clearly specify all u used, and the exact formula)
So I ask again, that there is somewhere only one singular definition of this concept, and a clear example how it is calculated – perhaps it would be best if the authors had a box with an example what is the contribution potential of one single paper, and what is the reproducible score/formula that anyone can use to get that number. As currently, it still escapes me how it was done.
2) Sentence: “Therefore, we aim to explore a use of 90 objective measurements for assessing peer review quality and their impact on paper contribution potential.” is not needed in the liter. review, you specify your aims in the introduction
3) Finally, the abstracts and discussion states: “We show that the exchanges between the
peer review committee and the authors, regardless of the number of revisions, implement a check for a baseline robustness of the final publication…. in response to peer reviewer feedback, authors add additional analysis to their papers” – I find these statements to be increadibly wrong, as there is no proof in this paper for them. Neither was robustness defined or measured, nor was adding of additional analyses – claiming that topic modeling proves this is completely wrong in my opinion as no comparison to manual detection of changes has ever been done. Therefore, authors should only report on associations they found and not make this kind of statements. In the rebuttal letter they stated this nicely: their results demonstrate that most revisions include a modification of these three topic categories, not of the actual changes.
4) In the rebuttal the authors said: We revised our manuscript so that the general section order and titles of the results section better reflects those of the methods. – and again, the order does not match, 4.1.1 and 5.1.1 are nor related - I would strongly encourage the editor to make the authors comply with this simple structuring request, and I will not bother with reviewing the rest, as it requires too much effort on my part again.

---

## Round 0.4 · accepted · Accept

I have reviewed your response to the reviewer who recommended rejection and am satisfied that you have now taken his suggestions seriously.